# The Protective Role of 28-Homobrassinolide and *Glomus versiforme* in Cucumber to Withstand Saline Stress

**DOI:** 10.3390/plants9010042

**Published:** 2019-12-26

**Authors:** Husain Ahmad, Sikandar Hayat, Muhammad Ali, Hongjiu Liu, Xuejin Chen, Jianming Li, Zhihui Cheng

**Affiliations:** 1College of Horticulture, Northwest A&F University, Yangling 712100, Shaanxi, China; husain@nwafu.edu.cn (H.A.); sikander@nwafu.edu.cn (S.H.); muhammadali@nwafu.edu.cn (M.A.); liured9@nwsuaf.edu.cn (H.L.); chenxuejin1982@126.com (X.C.); lijianming66@nwafu.edu.cn (J.L.); 2College of Natural Resources and Environment, Northwest A&F University, Yangling 712100, Shaanxi, China; 3College of Horticulture and Landscape Architecture, Henan Institute of Science and Technology, Xinxiang 453003, Henan, China

**Keywords:** arbuscular mycorrhizal fungi, 28-homobrassinolide, NaCl, antioxidant enzymes

## Abstract

The strategic role of phytohormones and arbuscular mycorrhizal fungi (AMF) to overcome various stress conditions is gaining popularity in sustainable agricultural practices. This current study aims to investigate and identify the protective roles of 28-homobrassinolide (HBL) and *Glomus versiforme* on two cucumber cultivars (salt sensitive Jinyou 1# and tolerant Chanchun mici (CCMC)) grown under saline conditions (100 mM NaCl). HBL and AMF were applied as individual and combined treatments on two cucumber cultivars and their effects were observed on the morphological growth and physiology under control and saline conditions. Findings revealed that the treated plants showed better performance under saline conditions through improved photosynthesis, leaf relative water content, and decreased electrolyte leakage in tolerant cultivar (CCMC) and to a lesser extent in sensitive (Jinyou 1#) cultivar. Comparable differences were noticed in the antioxidant enzymes activity such as superoxide dismutase, catalase, and peroxidase after every 10 days in both cultivars. Treating the plants with HBL and AMF also improved the mineral uptake regulation and lowered sodium concentration in roots compared to that in the non-treated plants. Current findings suggest that the protective role of HBL and AMF involves the regulation of antioxidants and lowers the risk of ion toxicity in the cucumber and hence enhance tolerance to salinity. These results are promising, but further studies are needed to verify the crop tolerance to stress and help in sustainable agricultural production, particularly vegetables that are prone to salinity.

## 1. Introduction

The world population is expected to exceed nine billion by the mid 21st century (Department of Economic and Social Affairs of the United Nations, 2015) and hence, food security will become a larger challenge. Therefore, sustainable food production is necessary to accommodate this augmentation in the population. There are several environmental obstacles (biotic and abiotic) to productivity among which, salinity is considered as the key menacing factor. The devastation of salinity has already affected about 7% of the world land [1,2,3], and it is considered as a key factor in increasing salinization. NaCl is responsible for hyper ionic and osmotic stress which causes overproduction of reactive oxygen species (ROS). Overproduction of ROS causes a decrease in photosynthesis, and damage to plant cells [4]. Plants scavenge ROS generated by salt stress through elevating antioxidant enzymes [2]. Various physiological and agronomic practices [5] are applied to minimize the damage due to salt stress, however, much progress has been made in determining the role of plant growth regulators in mediating physiological and biochemical responses which affect the growth and development of plants.

Brassinosteroids (BRs) are a class of phytohormones widely distributed in the plant kingdom that have a high potential for promoting growth and development [5,6,7,8,9]. To date, about 70 different forms of brassinosteroids have been isolated [9,10]. BRs have a critical role in root/stem growth, cell differentiation, cell elongation, gene modulation, development of flowers and fruits, etc., [6,7,9,10]. Reports also show that BRs regulate seed germination, maturation, senescence, and abscission [11,12]. Exogenous application of BRs influences ethylene biosynthesis by activating 1-aminocyclopropane-1-carboxylic acid synthesis pathway [13], elevating proton extrusion from membranes [14] and subsequently increasing production of nucleic acids and proteins [15]. Recently, BRs have been reported to ameliorate various stresses in plants [16]. Moreover, they are reported to reduce the influence of salt stress by enhancing nutrient uptake, improving mitotic index, growth [17,18], and activities of antioxidants preventing lipids degradation due to higher production of ROS [19].

Improving productivity under salinity is an important challenge and adopting new strategies to involve entities of biotic and abiotic nature would be of potential to cope with the excessive soil salinization. Plants are colonized by various microbes and mycorrhizal fungi which coordinate to involve in arbitrating various plant physiological functions especially nutrient uptake and tolerance to various stresses [20]. AMF (arbuscular mycorrhizal fungi) are symbionts that depend exclusively on the host and approximately ~20% of photosynthetic carbohydrates are obtained from the host [21]. They form a beneficial symbiotic relationship in the rhizosphere and play important roles in growth and root structure modification [21,22]. Despite AM fungi dependency on the host, in return, they adjust nutrients transfer and modulate plant growth by improving the water absorbance and nutrition by occupying additional area near the plant rhizosphere [23]. Hence AM fungi are directly related to the host plant’s physiology [2,24]. AMF is also responsible for ecosystem services, influencing the carbon and phosphorus cycle in a natural ecosystem, pest/stress resistance, and productivity [21,25,26]. The density of AM spores in salt affected soils are reported to be low, however, Aliasgharzadeh et al., and khan et al. [27,28] observed that the most predominant species of AMF in the severely saline soils of the Tabriz plains (with an electrical conductivity of around 160 dS m^−1^) were *Glomus intraradices*, *Glomus versiform*, and *Glomus etunicatum. G. versiforme* is reported to make improvements in net photosynthetic rate and water use efficiency [29]. Furthermore, AMF spp also elevates gas exchange in leaves [30,31], nutrient acquisition [32], rectified stomatal conductance [30,33], hydrolytic activity of roots [34,35], and scavenging of ROS through modulation of non-enzymatic and enzymatic antioxidants (superoxide dismutase (SOD), peroxidase (POD), ascorbate peroxidase (APX), catalase (CAT), etc.) under salinity [1,24,31]. Considering the potential roles of HBL (28-homobrassinolide) and AMF in ameliorating the stress conditions upon the receiver plants, and more importantly due to the lesser-known activity of these entities as a combined treatment, there is a dire need to investigate the strategic interaction of these two factors. The current study is therefore the first to investigate the combined application of HBL and AMF in the context of plant antioxidant enzymes, photosynthesis, and root activity of cucumber plants under saline stress and at various stages of plant growth.

## 2. Results

### 2.1. Plant Growth Attributes

Our results (Table 1) showed that NaCl reduced shoot and root length (68% and 65%), fresh weight (64% and 41%), and dry weight (50% and 23%) in cultivar Jinyou 1#, while it reduced shoot and root length (77% and 65%), fresh weight (60% and 40%), and dry weight (52% and 21%) in cultivar Chanchun mici (CCMC). The application of HBL and AMF improved growth attributes under salt stress. In addition, the combined effect of HBL and AMF caused an increase in the shoot and root length, fresh and dry weight in cultivar Jinyou 1# and shoot length, fresh and root dry weight in CCMC.

### 2.2. Chlorophyll and Root Activity

Salinity severely reduced chlorophyll a (75%), b (54%), a + b (50%), and root activity (51%) in cucumber cultivar Jinyou 1# and by 71%, 57%, 52%, and 50% in cultivar CCMC as compared to the control. According to Table 2, HBL and AMF showed an ameliorative effect in the stressed plants. The combined effect of HBL and AMF showed an increase in chlorophyll a, b, total chlorophyll, and root activity in Jinyou 1#, while the increase was not significant in cultivar CCMC.

### 2.3. Electrolyte Leakage and Leaf Relative Water Content

According to Figure 1, NaCl stress increased electrolyte leakage from cucumber leaves after 20 days by 174% and 135% in Jinyou 1# and CCMC, respectively. A similar trend was observed after 40 days when it was compared to their respective controls. Compared to stressed plants, HBL and AMF alone significantly decreased the electrolyte leakage of leaves. Moreover, the combination of AMF with HBL effectively reduced electrolyte leakage after 20 and 40 days by 32% and 40%, respectively in cultivar Jinyou 1# and 25% and 34% in cultivar CCMC, respectively.

The leaf relative water content (LRWC) showed a significant decline after 20 and 40 days of stress by 76% and 56% in cultivar Jinyou 1# and 72% and 52% in cultivar CCMC, respectively (Figure 1). The HBL foliar application and AMF root inoculation improved LRWC in cucumber leaves. Notwithstanding, the combined treatment significantly ameliorated LRWC only at 40 days in Jinyou 1#.

### 2.4. Photosynthetic Measurements

NaCl decreased photosynthetic activity by 58%, transpiration rate by 38%, stomatal conductance by 47%, and intercellular carbon dioxide concentration by 10% in the leaves of cultivar Jinyou 1# while it decreased photosynthetic activity by 54%, transpiration rate by 35%, stomatal conductance by 38%, and intercellular carbon dioxide concentration by 8% in CCMC. The HBL foliar application and AMF inoculation in roots (Figure 2) reduced the effects of NaCl stress mainly in the CCMC cultivar and to a lesser extent in Jinyou 1#. However, the combined effect of HBL and AMF showed higher results in the increased transpiration rate, intercellular CO_2_ concentration, photosynthesis, and stomatal conductance in both cultivars under stress conditions.

### 2.5. Colonization Percentage

As compared to non-stressed AMF inoculated plants, colonization percentage under salt stress was decreased by 51% and 46% in cultivar Jinyou 1# and CCMC, respectively. However, the combined effect of HBL and AMF inoculation under salt stress showed significant increments in colonization percentage by 10% and 15% compared to stressed plants in both cultivars (Figure 3).

### 2.6. Antioxidant Enzymes

#### 2.6.1. Superoxide Dismutase (SOD) Activity

The SOD activity of Jinyou 1# and CCMC was determined after every 10 days to understand its activity under prolonged stress conditions (Table 3 and Table 4). SOD activity showed increased increments after 10 days of stress, however, its activity decreased afterwards in NaCl stressed plants. The HBL foliar spray on plants under salt treatments elevated SOD activity after 10 and 20 days, while AMF continuously increased SOD activity after every 10 days under salt stress. Furthermore, the combined effect of HBL and AMF under salt treatments elevated the activity of SOD after every 10 days by 20%, 60%, 61%, and 72% in cultivar Jinyou 1# and 26%, 48%, 41%, and 60% in CCMC.

#### 2.6.2. Peroxidase (POD) Activity

Under salt stress, plants POD activity increased after 10, 20, and 30 days in cultivar Jinyou 1# and CCMC, however, it declined after 40 days in NaCl stressed plants (Table 3 and Table 4). The HBL foliar application under salt treatments increased POD activity after 10 and 20 days in both cultivars, while AMF alone showed gradual increment after every 10 days. The combined effect of HBL and AMF significantly increased POD activity by 65%, 71%, 65%, and 66% after every 10 days in cultivar Jinyou 1# and by 48%, 41%5, 45, and 43% in cultivar CCMC.

#### 2.6.3. Catalase (CAT) Activity

Catalase activity in plants under salt stress showed an increase after every 10 days by 49%, 83%, 86%, and 87% in cultivar Jinyou 1# and 33%, 49%, 63%, and 74% in CCMC. A significant trend was observed in the HBL foliar application and AMF inoculation after 30 and 40 days of salt treatments in both cultivars. Similarly, the combined effect of HBL and AMF significantly increased CAT activity after 30 days of salt treatments and the highest activity was noted after 40 days in both cultivars (Table 3 and Table 4).

#### 2.6.4. Malondialdehyde Content (MDA)

According to Table 3 and Table 4, the malondialdehyde content (MDA) in plants under salt stress continuously increased after every 10 days in both cultivars. Interestingly, the application of HBL and AMF resulted in a reduced MDA content as compared to their respective stress plants (without HBL and AMF). Nonetheless, the combined effect of HBL and AMF was clearly observed in reduced levels of MDA content after every 10 days by 5%, 18%, 9%, and 10% in Jinyou 1# and by 3%, 10%, 11%, and 8% in cultivar CCMC. These differences were not statistically significant except for cultivar CCMC after 20 and 30 days of NaCl.

### 2.7. Nitrogen, Phosphorous, Potassium, and Sodium

Salinity decreased nitrogen concentration in shoots by 59% and 70% in cultivar Jinyou 1# and CCMC however, there was no significant difference in roots (Table 5). Phosphorus concentration showed the same trend in both shoots (64% and 66%) and roots (85% and 67%) in both cultivars, whereas, potassium concentration increased in roots by 23% (Jinyou 1#) and 56% in CCMC and declined in shoots of both cultivars. Moreover, sodium ion concentration in shoots and roots showed increases in both cultivars. Although the HBL foliar application and AMF inoculation in roots revealed an ameliorative role in decreasing NaCl stress, their combination, however, showed prominent results in higher nitrogen, phosphorus, and potassium concentration in shoots and roots, while lower sodium concentration in shoots and roots of both cultivars, respectively.

## 3. Discussion

### 3.1. HBL, AMF, and Their Combined Effect on Growth and Biomass under Salt Stress

Salt stress severely affected the plant growth and AMF colonization in roots and biomass of both salt sensitive and salt tolerant cultivars. However, the cultivar CCMC exhibited more tolerance under severe conditions. The reduction of biomass in plants might be due to hyper ionic and osmotic stress produced by salinity [4] which causes disruption in cell organelles, photosynthesis, and respiration [36]. According to Hajiboland et al. [30], higher symbiosis was observed in tolerant varieties because their roots were less affected by NaCl stress. In our experiment, HBL, AMF, and their combination significantly enhanced growth and biomass as compared to their respective controls under salt stress. It has been reported that AMF, as compared to non-colonized plants, improved the growth and biomass of tomato [33,37], wheat, and pepper [34,38] under salt and water stress. Similarly, HBL has a positive effect on root and shoot growth and biomass in cucumber, *Vigna radiata,* and wheat under salt stress [6,7,39,40]. The increased growth attributes of cultivar CCMC might be due to its tolerant nature, enhanced nutrient acquisition through mycorrhizae [41] and positive effect on various osmolytes due to HBL [7], thereby increasing tolerance [42].

### 3.2. HBL, AMF, and Their Combined Influence on Chlorophyll Content, Root Activity, and Photosynthesis

Currents results showed a negative effect of NaCl on leaf chlorophyll content and root activity of both cultivars, however, the cultivar Jinyou 1# revealed less tolerance to chlorophyll degradation and photosynthesis which was in accordance with Hayat et al., and Ahmad et al. [7,39]. The decrease in chlorophyll content under salt stress might be due to overactivity of chlorophyllase, which damages chlorophyll content, and in turn, decreases photosynthetic activity and growth [37]. Reports have shown that salt stress has a deleterious impact on root structure [43]. The application of HBL, AMF, and their combination lowers the damage caused by salinity in cultivar CCMC. AMF inoculated plants have higher chlorophyll contents as compared to non-inoculated plants [23,37] and improved root activity [33]. Similar results were also observed in plants sprayed with HBL [6,7,44]. The higher root activity and chlorophyll contents of AMF inoculated plants can be attributed to a decrease in leaf sodium ion concentration and higher uptake of nutrients [37]. The HBL, on the other hand, enhances the activity of aquaporins through an increase in turgor pressure or proton pumping efficiency [45].

Plant photosynthesis, transpiration rate, and stomatal conductance were reduced in plants grown under salt stress, which corroborated with Hajiboland et al., and Ali et al. [30,46]. Leaf gaseous exchange in leaves occurs through stomata, and its function is severely altered due to higher concentration of sodium ions. The over reduction of ferrodoxin in the photosystem (I) lead to the production of O^−^_2_ by a process known as the Mehler reaction. This causes a chain reaction of reactive oxygen species (ROS) and leads to oxidative stress [2,47,48]. The HBL foliar application and root colonization with AMF, either alone or in combination, significantly improved gas exchange, transpiration rate, and photosynthesis in cultivar CCMC under salt stress. The higher gaseous exchange parameters in cultivar CCMC can be attributed to its tolerance nature as compared to Jinyou 1# which showed less tolerance under salt stress. HBL also ameliorates AMF colonization in the absence of NaCl. Hence, this is not a stress specific effect and HBL likely helps to cope with the stress by increasing AMF root colonization even without stress. It is documented that AMF causes a change in the physiology and structure of leaves and roots which result in elevated transpiration rate, photosynthesis, and gaseous exchange [22,30,49]. Similarly, HBL enhances absorption of nutrients and improves transfer of photosynthates from source to sink [46], and triggers the synthesis of specific proteins through the expression of genes that improve the metabolic activity of plants [50].

### 3.3. HBL, AMF, and Their Combined Effect on EC and LRWC

Current findings revealed that leaf EC and LRWC were adversely affected in plants under NaCl. However, cultivar Jinyou 1# was more affected as compared to cultivar CCMC. According to Ahmad et al., and Deinlein et al. [43,51] alteration in EC and LRWC is either due to oxidative stress or ionic toxicity. Sodium accumulation leads to oxidative stress and triggers the production of reactive oxygen species, which damage plant cells, organelles, and proteins. Moreover, alterations of the stomatal mechanisms and reduced water absorption result in the decline of LRWC and higher EC. The higher LRWC in leaves and lower EC in our study are possibly attributed to the enhanced hydrolytic activity of the roots, improved water, and nutrient absorption [41]. The higher water absorption also detoxifies ion toxicity and results in lower electrolyte leakage from leaves [22]. Similarly, HBL regulates cell division, elongation, and enhances efficiency of proton pumps to overcome the toxic effects of salts [46]. It also regulates water use efficiency and photosynthesis which in turn leads to the higher relative water content in leaves [5]. The cultivar CCMC retained higher LRWC and lower leakage of electrolytes under NaCl and hence their comparative levels in this regard may show its tolerant nature.

### 3.4. HBL, AMF, and Their Combined Effect on Changes of Antioxidant Enzymes

Changes were observed in antioxidant enzymes superoxide dismutase (SOD), peroxidase (POD) and catalase (CAT) whereas, malonaldehyde content (MDA), which is the end product of lipid peroxidation, dramatically increased in both cultivars due to saline stress conditions. Environmental stresses produce reactive oxygen species (ROS) in plants which lead to oxidative stress under salt stress [19,43]. The antioxidant enzymes, therefore, act as scavengers of these ROS in order to protect membranes, proteins of cellular compartments, from the devastation of stress conditions. It was observed that HBL and AMF application either alone or in combination, reduced lipid peroxidation under saline stress at a comparable degree to that of control plants. The antioxidant activity was relatively higher in CCMC cultivar as compared to Jinyou 1#. The increase in antioxidant activity removes the excessively produced ROS which can be attributed to the lower lipid peroxidation in plants under stress. These findings, therefore, elaborate the significant roles of antioxidant machinery activated due to HBL and AMF application which ultimately lowered the damage to cellular membranes possibly because of efficient ROS constitution [7,19,22]. Our results corroborated previous reports [7,46,52] which stated that HBL and AMF inoculation decreases lipid peroxidation under stress.

The stability of ROS production is dependent upon the activities of antioxidants. SOD has an affinity for superoxide radical and is the first line of defense to convert it to H_2_O and H_2_O_2_. POD and CAT further convert this H_2_O_2_ into H_2_O and O_2_ [47]. In our results, HBL and AMF application enhanced the antioxidant enzymes activity particularly in cultivar CCMC under stress conditions. AMF colonization has a role in enhancing tolerance of plants through triggering antioxidant enzymes by possessing several SOD genes under various stresses [53]. The enhanced activity of peroxidase and catalase in plants under AMF only are associated with decreased MDA content [30,49]. Similarly, HBL was reported to increase antioxidant levels in *Arabidopsis* plants, preventing them from oxidative stress [54]. Previous reports have shown that HBL modifies antioxidant enzymes and lowers membrane degradation [5,6,44,46,52,54] in heavy metal, salt, oxidative and temperature stress. HBL expressed genes encoding peroxidase *ATP2* and *ATP24*a in *Arabidopsis* [55] therefore, it is possible that HBL detoxifies ROS produced by facilitating peroxidase activity as part of a defense mechanism. The enhanced activity and biosynthesis of antioxidant enzymes in cultivar CCMC can be attributed to the increased transcription of genes encoding mRNA for antioxidant enzymes [10].

### 3.5. HBL, AMF, and Their Combined Effect on N, P, K, and Na^+1^ Ions

In our findings, plants grown under salt stress possessed higher Na ion accumulation and lower K, N, and P concentrations in both shoot and roots. In cultivar CCMC, a higher concentration of nitrogen, phosphorus, and potassium, while a relatively lower concentration of Na^+^ were observed in the shoot. Similarly, the same trend was observed in cultivar Jinyou 1#, but the K^+^/Na^+^ ratio was lower in both root and shoot as compared to cultivar CCMC, respectively. Plants under salt stress excessively accumulate Na^+^ which competes with K^+^ and disturbs ionic hemostasis in cells leading to lower uptake of mineral nutrients [56]. Ion toxicity can be attributed to the disruption of protein synthesis and enzyme activities, while, K^+^ plays a key role in balancing various physiological and biochemical changes in plants [57]. Under salt stress, essential nutrients uptake of phosphorus and nitrogen is hindered which leads to decreased production of proteins [18]. A higher Na^+^/K^+^ ratio due to salinity leads to low ion toxicity, and cell turgor pressure [22,57]. The lower Na^+^/K^+^ ratios in cultivar CCMC result in higher tolerance by reducing the uptake of sodium in higher concentrations. It has been reported that AMF alleviates the additional water stress through efficient water absorption by fungi hyphae, leading to a lower Na^+^/K^+^ ratio [1,30]. Phosphorus (P) is highly immobile in soil and reports have documented that AMF has a high affinity for phosphorus [58]. The higher nitrogen uptake in AMF inoculated plants might be due to the improved P uptake [59]. A recent study by Hammer et al. [60] stated that AMF can avoid uptake of Na^+^ ions by the selective uptake of potassium and calcium ions as osmotic equivalents. The Na^+^ might be stored in intraradical fungal hyphae, root cells, and vesicles to avoid allocation. AMF inoculated plants also show higher concentrations of magnesium and calcium including boron, iron, zinc, etc., [57]. Similarly, HBL reduces Na^+^/K^+^ ratio and improves nutrient concentrations [45]. Higher minerals uptake and improved proton pumps activity might be one of the mechanisms in stress tolerance in cultivar CCMC [7,10]. The increase in growth under HBL might be due to its part in ion hemostasis which is responsible for several physiological and metabolic functions in plants [7]. The higher tolerance of cucumber plants treated with the combination of HBL and AMF can be attributed to the increase in colonization of AM fungi in roots, phosphorus and potassium concentration in shoots, higher photosynthetic oxidative enzymatic activity (SOD and POD), and lower lipid peroxidation under salt stress.

## 4. Materials and Methods

### 4.1. Mycorrhizal Inoculums

The inoculum of arbuscular mycorrhizal fungus (*Glomus versiforme*) was received from the College of Horticulture, Northwest Agricultural and Forestry University (NWAFU), China. Maize plants were selected as host plants for the multiplication of AMF spores/hyphae under greenhouse conditions with optimum humidity, day, and night temperatures.

When this research was started, according to our knowledge, no specific work showed the optimum dose of HBL to be substantial at a particular stress level of NaCl. Therefore, we designed the experiments to determine and understand the adequate amount of HBL at variable stress levels of NaCl [36]. The arbuscular mycorrhizal (AM) *Glomus* species are widely distributed (regardless of the type and intensity of disturbance) in the ecosystem. Moreover, *G. versiforme* was already inoculated and studied extensively at the college of horticulture and forestry (NWAFU). As stated, less was known about the role of HBL in ameliorating the salinity effects on growth of cucumber. Therefore, two different cultivars of cucumber were used in our study to elaborate the effectiveness of HBL per se in the salinity stress situations.

### 4.2. Plant Material and Mycorrhizal Fungus Inoculums

Cucumber seeds, salt sensitive (Jinyou 1#), and tolerant Chanchun mici (CCMC) were surface sterilized with 0.5% NaOCl solution and washed thoroughly afterwards. Then, the cucumber plant seed nursery was grown in small plastic trays, half of the plant seeds received arbuscular mycorrhizal fungi inoculated soil for better colonization. Twenty-day-old uniform seedlings were transplanted to 300 × 300 mm pots under a plastic tunnel. Pots containing AMF received 11 g of AMF inoculum per 0.85 kg of soil adjacent to the seedling roots. The non-inoculated plants received autoclaved inoculum with microbial culture filtrate. Each pot consisted of 6 kg of soil consisting of soil, sand, and organic matter (*v*/*v* 2:1:1). Soil properties were pH 7.57, 0.485 g kg^−1^ total nitrogen, 0.111 g kg^−1^ phosphorus, 0.272 g kg^−1^ potassium, and 11 g kg^−1^ organic matter.

### 4.3. Pot Experiment

A stock solution of HBL 1 milli Molar was initially prepared by using a little amount of pure ethanol as a solvent. Stock solution was further diluted for the desired 1 µM concentration. Plant leaves were sprayed the day after NaCl application. Sodium chloride at 100 mM (EC 10.56 ds m^−1^) concentration was applied by irrigation. Pots’ EC values were kept constant. There was a total of eight treatments: (1) Control, (2) HBL spray, (3) AMF, (4) HBL + AMF, (5) NaCl, (6) HBL + NaCl, (7) AMF + NaCl, and (8) HBL + AMF + NaCl, and each treatment consisted of 10 pots per replication. The experiment was replicated three times, and a total of 240 pots were used for a single cultivar. The selected treatments of NaCl were selected based on our previous research [39].

### 4.4. Plant Growth Attributes

Plant growth attributes were recorded after 40 days of treatments. Length and fresh mass were recorded using a measuring tape and weight balance and dry weight was determined by drying the materials in the oven for 72 h at 80 °C [6].

### 4.5. Chlorophyll Content

Leaf chlorophyll content was determined in fresh leaves of 200 mg (40 days after germination) extracted in 80% acetone as determined by [61]. The absorbance was noted through spectrophotometer (UV-4802, UNICO, MDN, USA).

### 4.6. Root Activity

Plant roots of 0.5 g were taken, and root activity was determined through triphenyl-tetrazolium chloride (TTC) [62]. The extract absorbance was measured at 485 nm wavelength by a spectrophotometer. The root activity was expressed as TTC reducing intensity (mg g^−1^ h^−1^) and measured as per linear equation from the following formula:(1)$$\mathrm{TTC}\;\left(\mathrm{mg}/g\cdot h\right)=\frac{\mathrm{Reduction}\;\mathrm{mass}\;\left(\mathrm{mg}\right)\;\mathrm{of}\;\mathrm{TTC}}{\mathrm{Root}\;\mathrm{weight}\;\left(\mathrm{mg}\right)}\times \;\mathrm{time}\;\left(h\right)\;\times \;100$$

### 4.7. Electrolyte Leakage (%)

The leaf electrolyte leakage was determined according to [63]. EC_a_ was noted by taking 20 leaf discs per treatment (20 and 40 days after salt treatments). The tubes were heated in a water bath for 30 min at 50 °C and EC_b_ was measured. The EC_c_ was noted after boiling the samples at 100 °C for 10 min. The electrolyte leakage was determined as follows:EC (%) = ((EC_b_ − EC_a_)/(EC_c_)) × 100(2)

### 4.8. Relative Leaf Water Content (%)

Leaf relative water content was determined by Smart et al. [64]. Fresh leaf discs weighted (after 20 and 40 days of salt treatments) were placed in petri dishes with distilled water for 24 h and turgor mass was noted. The samples were kept in the oven for 24 h at 80 °C and dry mass was measured.
Relative leaf water content (RLWC) (%) = ((Fresh mass − Dry mass)/(Turgor mass − Dry mass)) × 100(3)

### 4.9. Photosynthetic Measurements

The third leaf from the top was used for the measurement of stomatal conductance, intercellular CO_2_ concentration, transpiration rate, and net photosynthesis rate through portable photosynthetic system (LI-COR 6400XT) from 10:00 a.m. to 12:00 p.m. on a sunny day after 40 days of NaCl application.

### 4.10. AMF Colonization

At the end of the experiment, fine roots were collected from treatments in replicate. Root samples (1 cm in length) were treated with hydrochloric acid and potassium hydroxide followed by staining with trypan blue [65]. Samples were magnified under a light microscope (Olympus, Tokyo, Japan). Fifty root pieces per treatment were used to calculate the colonization percentage as determined by [66]. The colonization was calculated as follows:(4)$$\mathrm{AMF}\%=\frac{\mathrm{Roots}\;\mathrm{pieces}\;\mathrm{having}\;\mathrm{AMF}}{\mathrm{Roots}\;\mathrm{pieces}\;\mathrm{observed}}\;\times \;100$$

### 4.11. Antioxidant Enzymes

Then, 0.5 g of healthy leaf samples was grounded in liquid nitrogen, homogenized in buffer (phosphate buffer 0.05 M, pH 7.8), and centrifuged at 12,000× *g* at a temperature of 4 °C for 20 min. The supernatant was for the deduction of SOD, POD, and CAT) activities, and MDA content.

Total SOD activity (superoxide dismutase; EC 1.15.1.1) was determined through nitro blue tetrazolium (NBT) [67]. The reaction mixture was subjected to fluorescent light (85.85 µmol m^−2^ s^−1^) exposure for 20 min and color absorbance was measured on a spectrophotometer at 560 nm wavelength. POD (peroxidase; EC 1.11.1.7) activity was observed as determined by [68]. The absorbance (ug-g^−1^ FW min^−1^) was observed at 470 nm at 30 s interval for 3 min. CAT (catalase; EC 1.11.1.6) activity was assayed by measuring H_2_O_2_ reduction [69]. The hydrogen peroxide reduction at 240 nm wavelength was noted after every 30 s till 3 min. The MDA content was determined through thiobarbituric acid (TBA) reaction method [70]. Then, 2 mL TBA solution and 1 mL enzyme extract was dissolved in trichloroacetic acid (TCA) 5% (*v*/*v*) and heated in a water bath for 15 min. The absorbance was noted at 450, 532, and 600 nm.

### 4.12. Nutrients Determination in Plants

Plant shoot and root samples (0.5 g) were oven dried, grounded, and digested. Distilled water was used to raise the volume up to 100 mL. Nitrogen (N) was measured by a modified micro-Kjeldahl method [71]. Sodium (Na^+^) and potassium (K^+^) were measured through a flame photometer, while phosphorous (P) was measured through a spectrophotometer [72].

### 4.13. Statistical Analysis

The experiment was designed according to the split plot arrangement. The main plot consisted of sodium chloride (100 mM) while, HBL and AMF treatments were a subplot with three times replicate. The significant differences in means between the treatments were separated according to the least significant difference (probability, *p* < 0.05) by using SPSS statistics 17.0. The analysis of variance for each parameter is presented in Appendix A for cultivar Jinyou 1# and Appendix A for cultivar CCMC respectively in the Appendix A.

## 5. Conclusions

Based on our findings, the improved growth attributes and nutrition in AMF colonized plants might be due to its involvement in nutrient accumulation and additional water absorption through its hyphae. On the other hand, the improved photosynthesis, gaseous parameters, and chlorophyll content suggest the participation of HBL, which leads to the elevated growth of cucumber plants under salt stress. Findings of current research provide a significant platform to investigate the mechanism of AMF + HBL in improving plant growth, biomass, photosynthetic parameters, ion hemostasis in cells, electrolyte leakage, the antioxidant system, and improved uptake of nutrients under adverse conditions. Nonetheless, HBL and AMF combination can be formulated to increase vegetable production, particularly cucumber under saline conditions.

## Figures and Tables

**Figure 1 plants-09-00042-f001:**
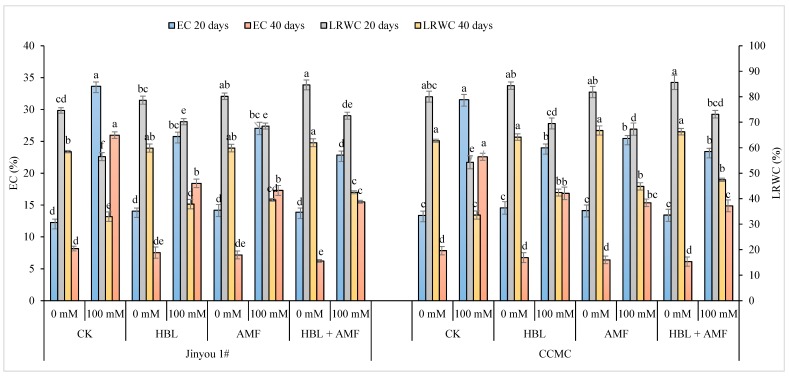
HBL, AMF, and their combination on NaCl (100 mM) induced changes on electrolyte leakage (EC %) and leaf relative water content (LRWC %) after 20 and 40 days (flowering and fruiting stage) of NaCl treatment on cucumber cultivars Jinyou 1# (salt sensitive) and CCMC (salt tolerant). Values presented for comparison are means of three replications in each treatment (±SE, standard error). Means followed by the same letters are not significantly different (*p* ≤ 0.05) according to the least significant difference (LSD) having three replications.

**Figure 2 plants-09-00042-f002:**
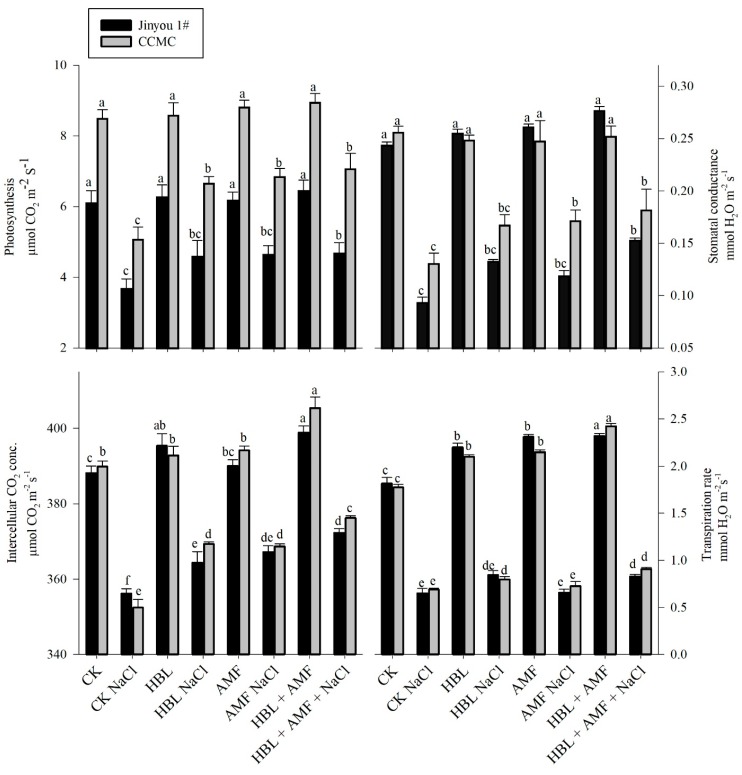
HBL, AMF, and their combination on the NaCl (100 mM) induced changes on photosynthesis (mmol CO_2_ m^−2^ S^−1^), intercellular carbon dioxide concentration (µmol CO_2_ m^−2^ S^−1^), stomatal conductance (mmol H2O m^−2^ s^−1^), and transpiration rate (mmol H_2_O m^−2^ s^−1^) of cucumber cultivars (after 40 days NaCl treatment, fruiting stage), Jinyou 1# (salt sensitive) and CCMC (salt tolerant). Values presented for comparison are means of three replications in each treatment (±SE, standard error). Means followed by the same letters are not significantly different *(p ≤* 0.05) according to the least significant difference (LSD) having three replications.

**Figure 3 plants-09-00042-f003:**
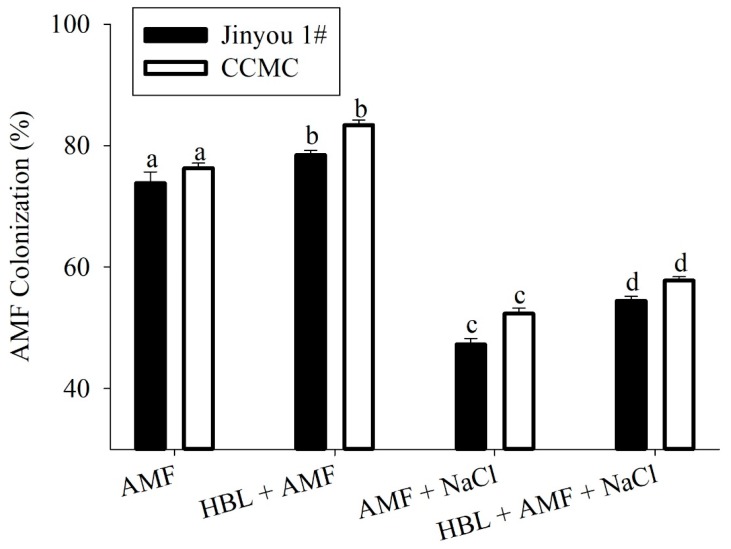
HBL, AMF, and their combination on the NaCl (100 mM) induced changes on colonization percentage (%) of AMF inoculum in roots of cucumber cultivars, Jinyou 1# (salt sensitive) and CCMC (salt tolerant). Colonization percentage was determined at the end of the experiment (after 40 days, fruiting stage). Values presented for comparison are means of three replications in each treatment (±SE, standard error). Values are means of at least three replications (±SE, standard error) presented for comparison. Means followed by the same letters are not significantly different (*p ≤* 0.05) according to the least significant difference (LSD) having three replications.

**Table 1 plants-09-00042-t001:** Effect of HBL, AMF, and their combined application on the phenotype of cucumber cultivars grown under saline condition.

Treatments		Shoot Length (cm)	Root Length (cm)	Shoot Fresh Weight (g)	Shoot Dry Weight (g)	Root Fresh Weight (g)	Root Dry Weight (g)
Jinyou 1# (salt sensitive)							
CK	0 mM	94.8 ± 1.6 b	53.4 ± 2.7 a	101.0 ± 4.0 b	15.7 ± 1.9 ab	18.6 ± 0.8 ab	3.0 ± 0.1 a
	100 mM	64.5 ± 1.2 d	35.1 ± 2.1 c	65.2 ± 4.1 d	7.8 ± 1.5 d	7.7 ± 1.3 d	0.7 ± 0.1 c
HBL	0 mM	97.6 ± 2.1 ab	54.1 ± 1.6 a	110.6 ± 4.5 b	16.1 ± 2.3 ab	19.1 ± 0.5 ab	3.0 ± 0.2 a
	100 mM	72.4 ± 2.9 c	42.8 ± 1.3 b	77.5 ± 3.6 cd	10.4 ± 1.0 cd	11.8 ± 1.5 c	1.2 ± 0.2 bc
AMF	0 mM	99.3 ± 2.6 ab	53.1 ± 1.0 a	129.9 ± 4.0 a	17.9 ± 1.3 a	20.1 ± 1.2 a	3.4 ± 0.2 a
	100 mM	73.6 ± 2.1 c	43.8 ± 2.1 b	68.1 ± 4.7 cd	11.4 ± 0.7 cd	12.2 ± 1.6 c	1.3 ± 0.3 bc
HBL + AMF	0 mM	101.5 ± 2.8 a	54.6 ± 1.8 a	135.2 ± 3.1 a	18.9 ± 1.0 a	21.0 ± 1.1 a	3.8 ± 0.3 a
	100 mM	75.4 ± 1.5 c	45.8 ± 1.7 b	81.1 ± 5.2 c	12.4 ± 0.5 bc	14.2 ± 1.1 bc	1.7 ± 0.3 b
CCMC (salt tolerant)							
CK	0 mM	84.4 ± 1.5 ab	48.7 ± 2.9 abc	101.0 ± 2.5 b	16.7 ± 2.1 abc	17.8 ± 2.5 abc	3.7 ± 0.1 b
	100 mM	65.1 ± 1.5 d	32.5 ± 2.3 d	60.9 ± 3.1 d	8.7 ± 1.5 d	10.7 ± 1.2 d	0.8 ± 0.1 d
HBL	0 mM	87.9 ± 2.9 a	49.4 ± 3.7 abc	103.7 ± 1.2 ab	17.2 ± 2.3 abc	19.31 ± 3.1 abc	3.9 ± 0.2 ab
	100 mM	75.0 ± 2.0 c	37.1 ± 3.1 d	79.6 ± 2.3 c	12.1 ± 1.4 cd	12.9 ± 1.3 cd	2.1 ± 0.2 c
AMF	0 mM	89.4 ± 2.9 a	51.9 ± 3.5 ab	106.5 ± 3.8 ab	19.0 ± 2.0 ab	20.2 ± 1.6 ab	4.3 ± 0.1 ab
	100 mM	76.9 ± 1.1 bc	38.5 ± 3.9 cd	78.9 ± 3.1 c	12.3 ± 0.7 cd	14.7 ± 1.2 bcd	2.2 ± 0.3 c
HBL + AMF	0 mM	90.4 ± 2.8 a	53.3 ± 3.9 a	108.0 ± 1.4 a	21.9 ± 0.7 a	22.1 ± 1.5 a	4.4 ± 0.1 a
	100 mM	78.2 ± 2.4 bc	40.7 ± 2.8 bcd	83.3 ± 2.2 c	13.0 ± 1.5 bc	15.0 ± 1.5 bcd	2.4 ± 0.1 c

Data were collected for cucumber genotypes (Jinyou 1# and Chanchun mici (CCMC)) 40 days (i.e., fruiting stage) after treatment with NaCl. Values presented for comparison are means of three replications in each treatment (±SE, standard error). Means followed by the same letters are not significantly different (*p* ≤ 0.05) according to the least significant difference (LSD) having three replications. HBL, 28-homobrassinolide; AMF, arbuscular mycorrhizal fungi; CK, control.

**Table 2 plants-09-00042-t002:** Effect of HBL, AMF, and their combined application on the chlorophyll contents and root activity of cucumber cultivars grown under saline condition.

Treatments		Chlorophyll a (mg g^−1^ FW)	Chlorophyll b (mg g^−1^ FW)	Chlorophyll a + b (mg g^−1^ FW)	Root Activity (mg g^−1^ h^−1^)
Jinyou 1# (salt sensitive)					
CK	0 mM	16.5 ± 0.2 bc	4.3 ± 0.2 a	20.9 ± 0.2 c	21.8 ± 0.5 a
	100 mM	12.3 ± 0.1 e	2.3 ± 0.1 d	14.7 ± 0.1 f	10.8 ± 0.1 d
HBL	0 mM	17.1 ± 0.1 ab	4.4 ± 0.1 a	21.6 ± 0.1 ab	22.0 ± 0.8 a
	100 mM	15.1 ± 0.2 d	3.4 ± 0.1 bc	18.6 ± 0.3 e	13.1 ± 0.2 c
AMF	0 mM	16.6 ± 0.1 bc	4.4 ± 0.1 a	21.0 ± 0.2 bc	22.6 ± 0.2 a
	100 mM	15.2 ± 0.3 d	3.1 ± 0.1 c	18.4 ± 0.3 e	14.0 ± 0.5 bc
HBL + AMF	0 mM	17.4 ± 0.2 a	4.5 ± 0.1 a	22.0 ± 0.1 a	23.5 ± 0.3 a
	100 mM	15.7 ± 0.1 cd	3.5 ± 0.1 b	19.3 ± 0.1 d	15.2 ± 0.4 b
CCMC (salt tolerant)					
CK	0 mM	17.3 ± 0.1 a	4.9 ± 0.4 a	22.3 ± 0.5 a	25.5 ± 1.3 a
	100 mM	12.4 ± 0.3 c	2.8 ± 0.2 b	15.3 ± 0.1 c	13.0 ± 0.7 c
HBL	0 mM	17.4 ± 0.1 a	5.3 ± 0.2 a	22.8 ± 0.3 a	26.1 ± 1.3 a
	100 mM	14.7 ± 0.1 b	4.1 ± 0.5 ab	18.9 ± 0.6 b	18.2 ± 1.4 b
AMF	0 mM	17.4 ± 0.1 a	5.1 ± 0.3 a	22.5 ± 0.2 a	26.8 ± 1.1 a
	100 mM	14.9 ± 0.2 b	4.1 ± 0.5 ab	19.0 ± 0.5 b	19.1 ± 0.6 b
HBL + AMF	0 mM	17.5 ± 0.1 a	5.6 ± 0.2 a	23.1 ± 0.1 a	28.1 ± 0.4 a
	100 mM	15.2 ± 0.1 b	4.5 ± 0.2 ab	19.8 ± 0.3 b	21.1 ± 0.2 b

Data were collected for cucumber genotypes (Jinyou 1# and CCMC) 40 days (i.e., fruiting stage) after treatment with NaCl. Values presented for comparison are means of three replications in each treatment (±SE, standard error). Means followed by the same letters are not significantly different (*p* ≤ 0.05) according to the least significant difference (LSD) having three replications. FW, fresh weight.

**Table 3 plants-09-00042-t003:** HBL, AMF, and their combination on the NaCl (100 mM) induced changes on superoxide dismutase (SOD, µg^−1^ FW h^−1^), peroxidase (POD, µg-g^−1^ FW min^−1^), catalase (CAT, µg-g^−1^ FW min^−1^), and malondialdehyde content (MDA, mmol g^−1^ FW).

Cultivar	Treatments		SOD (µg^−1^ FW h^−1^)			
			10 Days	20 Days	30 Days	40 Days
Jinyou 1# (salt sensitive)	CK	0 mM	333.5 ± 13.3 cd	349.5 ± 15.3 c	345.7 ± 17.4 c	350.3 ± 18.1 c
		100 mM	467.7 ± 28.1 a	605.2 ± 18.2 a	587.3 ± 23.1 ab	514.6 ± 18.9 b
	HBL	0 mM	324.8 ± 16.9 cd	357.6 ± 21.9 c	351.6 ± 19.2 c	348.1 ± 25.1 c
		100 mM	469.6 ± 23.2 a	589.6 ± 14.3 ab	627.8 ± 12.4 a	559.5 ± 15.5 ab
	AMF	0 mM	319.9 ± 15.8 d	341.1 ± 24.8 c	342.6 ± 24.3 c	347.2 ± 23.7 c
		100 mM	382.1 ± 17.5 bc	526.2 ± 25.5 b	565.6 ± 17.5 b	592.3 ± 20.7 a
	HBL + AMF	0 mM	332.8 ± 21.3 cd	344.8 ± 21.3 c	344.4 ± 18.8 c	353.8 ± 20.6 c
		100 mM	401.3 ± 20.2 b	565.3 ± 22.3 b	554.6 ± 23.5 b	604.1 ± 23.7 a
			POD (µg-g^−1^ FW min^−1^)			
			10 days	20 days	30 days	40 days
	CK	0 mM	706.1 ± 25.2 d	738.5 ± 18.5 c	759.6 ± 22.5 c	774.6 ± 18.6 c
		100 mM	1093.8 ± 22.7 c	1177.1 ± 24.8 b	1184.1 ± 20.6 b	1168.1 ± 24.1 b
	HBL	0 mM	742.5 ± 15.3 d	754.7 ± 22.8 c	748.9 ± 18.2 c	782.5 ± 19.3 c
		100 mM	1230.6 ± 22.8 a	1304.3 ± 16.4 a	1237.9 ± 22.4 ab	1249.2 ± 14.7 a
	AMF	0 mM	724.2 ± 21.6 d	768.1 ± 12.3 c	773.6 ± 20.7 c	801.8 ± 17.1 c
		100 mM	1132 ± 12.4 ac	1190.3 ± 19.7 b	1178.4 ± 24.9 b	1262.2 ± 16.2 a
	HBL + AMF	0 mM	764.2 ± 24.7 d	772.5 ± 10.4 c	783.8 ± 22.2 c	811.1 ± 14.4 c
		100 mM	1171 ± 18.4 ab	1262.6 ± 25.7 a	1263.4 ± 17.1 a	1292.6 ± 16.4 a
			CAT (µg-g^−1^ FW min^−1^)			
			10 days	20 days	30 days	40 days
	CK	0 mM	327.8 ± 26.1 b	302.9 ± 14.2 b	317.7 ± 18.7 c	326.5 ± 19.6 c
		100 mM	488.3 ± 21.6 a	553.2 ± 14.4 a	593.9 ± 13.1 b	620.9 ± 17.3 b
	HBL	0 mM	341 ± 28.8 b	335.8 ± 14.5 b	322.3 ± 16.4 c	332.2 ± 18.5 c
		100 mM	509.6 ± 17.4 a	588.1 ± 25.1 a	602.5 ± 14.4 ab	665.8 ± 22.7 ab
	AMF	0 mM	348.4 ± 24.9 b	346.8 ± 21.9 b	336.1 ± 17.6 c	342.7 ± 23.3 c
		100 mM	475.9 ± 29.6 a	567.2 ± 15.8 a	616.2 ± 15.2 ab	678.1 ± 19.6 ab
	HBL + AMF	0 mM	349.4 ± 26.7 b	352.4 ± 23 b	339.8 ± 14.6 c	348.5 ± 14.5 c
		100 mM	499.4 ± 18.1 a	562.4 ± 22.1 a	625.4 ± 19.3 a	722.9 ± 18.1 a
			MDA (mmol g^−1^ FW)			
			10 days	20 days	30 days	40 days
	CK	0 mM	1.03 ± 0.14 c	1.66 ± 0.56 c	1.63 ± 0.18 c	1.72 ± 0.07 c
		100 mM	1.89 ± 0.26 a	2.95 ± 0.40 a	3.72 ± 0.20 a	4.26 ± 0.20 a
	HBL	0 mM	1.08 ± 0.18 c	1.48 ± 0.44 c	1.51 ± 0.22 c	1.64 ± 0.17 c
		100 mM	1.77 ± 0.17 ab	2.41 ± 0.51 ab	3.27 ± 0.26 ab	4.11 ± 0.24 ab
	AMF	0 mM	1.11 ± 0.10 c	1.54 ± 0.36 c	1.59 ± 0.21 c	1.63 ± 0.07 c
		100 mM	1.82 ± 0.13 ab	2.44 ± 0.37 ab	3.30 ± 0.27 ab	3.91 ± 0.15 ab
	HBL + AMF	0 mM	1.18 ± 0.05 c	1.64 ± 0.30 c	1.50 ± 0.22 c	1.61 ± 0.16 c
		100 mM	1.81 ± 0.16 b	2.4 ± 0.49 b	3.39 ± 0.27 b	3.85 ± 0.15 b

Data were collected for cucumber genotype (Jinyou 1#) 10, 20, 30, and 40 days (i.e., vegetative, flowering, and fruiting stage) after treatment with NaCl. Values presented for comparison are means of three replications in each treatment (±SE, standard error). Means followed by the same letters are not significantly different *(p ≤* 0.05) according to the least significant difference (LSD) having three replications.

**Table 4 plants-09-00042-t004:** HBL, AMF, and their combination on the NaCl (100 mM) induced changes on superoxide dismutase (SOD, µg^−1^ FW h^−1^), peroxidase (POD, µg-g^−1^ FW min^−1^), catalase (CAT, µg-g^−1^ FW min^−1^), and malondialdehyde content (MDA, mmol g^−1^ FW).

Cultivar	NaCl (100 mM)	Treatments	SOD (µg^−1^ FW h^−1^)			
			10 Days	20 Days	30 Days	40 Days
CCMC (salt tolerant)	CK	0 mM	498.2 ± 18.2 c	511.2 ± 20.6 c	534.5 ± 19.5 d	519.8 ± 18.7 e
		100 mM	669.7 ± 19.8 ab	788.6 ± 23.2 ab	679.4 ± 15.3 c	730.4 ± 23.1 c
	HBL	0 mM	503.8 ± 23.5 c	513.6 ± 19.3 c	567.1 ± 17 d	525.4 ± 15.3 d
		100 mM	697.8 ± 20.8 a	806.9 ± 27.4 a	723.9 ± 13.3 bc	754.2 ± 17 b
	AMF	0 mM	499.6 ± 25.5 c	506.3 ± 21.9 c	563.8 ± 13.9 d	527.7 ± 16.3 d
		100 mM	671.7 ± 20.6 ab	734.8 ± 20.1 b	732.5 ± 20.9 b	778.6 ± 17.3 ab
	HBL + AMF	0 mM	504.7 ± 23.5 c	517.7 ± 20.9 c	569.6 ± 16.1 d	528.6 ± 11.2 d
		100 mM	632.9 ± 26.2 b	760.3 ± 19.9 ab	753.5 ± 18.9 a	827.7 ± 12.7 a
			POD (µg-g^−1^ FW min^−1^)			
			10 days	20 days	30 days	40 days
	CK	0 mM	930.4 ± 21.6 c	976.2 ± 15.7 c	981.1 ± 15.9 c	997.6 ± 16.5 d
		100 mM	1311.1 ± 29.7 b	1332.2 ± 23.4 b	1359.5 ± 12.3 b	1318.7 ± 13.5 c
	HBL	0 mM	944.9 ± 22.4 c	981.4 ± 21.4 c	986.2 ± 16.8 c	1002.2 ± 14.5 d
		100 mM	1401.5 ± 23.1 a	1427.3 ± 24.8 a	1398.1 ± 21.4 ab	1373.1 ± 15.3 b
	AMF	0 mM	948.4 ± 27.9 c	988.2 ± 22.6 c	993.2 ± 20.4 c	1009.7 ± 14.3 d
		100 mM	1342.6 ± 28.3 ab	1367.2 ± 24.5 ab	1409.9 ± 19.9 ab	1415.1 ± 15.6 ab
	HBL + AMF	0 mM	956.2 ± 19.5 c	996.2 ± 19.1 c	997.5 ± 17.1 c	1018.5 ± 18.4 d
		100 mM	1382.7 ± 18.2 a	1384.2 ± 25.1 ab	1429.8 ± 18.8 a	1433.3 ± 19.6 a
			CAT (µg-g^−1^ FW min^−1^)			
			10 days	20 days	30 days	40 days
	CK	0 mM	468.3 ± 22.5 b	488.5 ± 20.3 b	498.7 ± 16.1 c	508.2 ± 11.7 d
		100 mM	623.6 ± 27.2 a	729.2 ± 22.5 a	761.6 ± 19.8 a	845.3 ± 15.6 c
	HBL	0 mM	516.5 ± 33.3 b	504.3 ± 18.9 b	523.1 ± 23.6 c	517.6 ± 15.6 d
		100 mM	649.3 ± 27.6 a	756.8 ± 13.2 a	819.4 ± 23.7 b	891.4 ± 18.1 b
	AMF	0 mM	502.1 ± 24.3 b	525.8 ± 21.1 b	529.8 ± 20.7 c	539.1 ± 15.2 d
		100 mM	641.6 ± 34.1 a	718.3 ± 12.8 a	831.1 ± 21.8 b	903.3 ± 20.8 ab
	HBL + AMF	0 mM	510.9 ± 11.9 b	536.2 ± 15.5 b	535.9 ± 23.3 c	544.7 ± 21.2 d
		100 mM	653.9 ± 34.2 a	746.7 ± 11.9 a	863.7 ± 21.4 b	936.8 ± 14.3 a
			MDA (mmol g^−1^ FW)			
			10 days	20 days	30 days	40 days
	CK	0 mM	1.84 ± 0.09 c	2.21 ± 0.09 c	2.45 ± 0.04 c	2.60 ± 0.01 c
		100 mM	2.52 ± 0.09 a	3.72 ± 0.06 a	4.37 ± 0.16 a	4.93 ± 0.13 a
	HBL	0 mM	1.88 ± 0.05 c	2.23 ± 0.11 c	2.42 ± 0.03 c	2.56 ± 0.06 c
		100 mM	2.41 ± 0.06 ab	3.31 ± 0.08 b	3.98 ± 0.07 b	4.70 ± 0.13 ab
	AMF	0 mM	1.90 ± 0.04 c	2.25 ± 0.09 c	2.41 ± 0.02 c	2.55 ± 0.06 c
		100 mM	2.45 ± 0.05 ab	3.37 ± 0.07 b	3.87 ± 0.12 b	4.60 ± 0.15 ab
	HBL + AMF	0 mM	1.90 ± 0.06 c	2.17 ± 0.10 c	2.38 ± 0.10 c	2.54 ± 0.11 c
		100 mM	2.36 ± 0.08 b	3.34 ± 0.06 b	3.97 ± 0.09 b	4.57 ± 0.13 b

Data were collected for cucumber genotype (CCMC) 10, 20, 30, and 40 days (i.e., vegetative, flowering, and fruiting stage) after treatment with NaCl. Values presented for comparison are means of three replications in each treatment (±SE, standard error). Means followed by the same letters are not significantly different *(p ≤* 0.05) according to the least significant difference (LSD) having three replications.

**Table 5 plants-09-00042-t005:** HBL, AMF, and their combination on the NaCl (100 mM) induced changes in shoot and root concentration of N, P^+^, K^+^, and Na^+^ in cucumber.

NaCl (100 mM)	Treatments	Nitrogen (µg/L)		Phosphorus (µg/L)		Potassium (µg/L)		Sodium (µg/L)		K/Na ratio	
		Shoot	Root	Shoot	Root	Shoot	Root	Shoot	Root	Shoot	Root
Jinyou 1# (salt sensitive)											
CK	0 mM	48.7 ± 2.1 a	16.6 ± 2.6 a	43.8 ± 0.8 a	98.8 ± 1.0 c	84.1 ± 1.2 b	86.6 ± 1.1 e	23.7 ± 0.8 d	29.7 ± 2.5 d	3.5 ± 0.1 b	2.9 ± 0.2 c
	100 mM	29.3 ± 0.5 d	12.2 ± 2.7 a	28.1 ± 0.6 d	83.6 ± 2.4 e	50.4 ± 1.1 e	139.7 ± 1.3 a	156.5 ± 1.3 a	149.6 ± 2.5 a	0.3 ± 0.1 c	0.9 ± 0.1 d
HBL	0 mM	49.3 ± 3.3 a	21.2 ± 3.7 a	44.8 ± 1.0 a	105.7 ± 0.4 b	86.1 ± 0.8 b	92.3 ± 1.1 de	22.9 ± 1.3 d	24.5 ± 2.1 d	3.7 ± 0.2 b	3.8 ± 0.3 bc
	100 mM	36.3 ± 0.8 c	14.3 ± 2.0 a	31.5 ± 1.5 cd	87.4 ± 2.6 de	57.9 ± 1.6 d	123.1 ± 1.2 b	90.3 ± 1.7 b	127.4 ± 3.1 b	0.6 ± 0.1 c	0.9 ± 0.3 d
AMF	0 mM	52.1 ± 2.3 a	18.7 ± 2.5 a	46.6 ± 0.5 a	110.9 ± 2.0 b	89.1 ± 1.7 ab	96.6 ± 1.0 cd	19.8 ± 0.8 d	24.3 ± 2.0 d	4.5 ± 0.1 a	4.0 ± 0.3 b
	100 mM	38.1 ± 1.9 bc	14.9 ± 3.7 a	33.9 ± 1.8 bc	91.7 ± 2.6 cd	61.9 ± 1.3 d	116.3 ± 1.1 b	88.1 ± 1.5 bc	113.0 ± 2.8 c	0.7 ± 0.1 c	1.0 ± 0.3 d
HBL + AMF	0 mM	53.7 ± 1.5 a	24.4 ± 3.8 a	47.2 ± 1.4 a	118.9 ±1.7 a	93.2 ± 1.7 a	101.5 ± 1.8 c	19.5 ± 1.6 d	20.1 ± 1.9 d	4.8 ± 0.1 a	5.1 ± 0.5 a
	100 mM	42.7 ± 0.6 b	16.3 ± 3.1 a	35.7 ± 0.2 b	93.3 ± 1.1 c	67.9 ± 1.3 c	106.7 ± 1.5 c	82.6 ± 0.6 c	103.4 ± 2.3 c	0.8 ± 0.1 c	1.0 ± 0.3 d
CCMC (salt tolerant)											
CK	0 mM	44.8 ± 1.3 ab	10.2 ± 0.9 a	47.9 ± 1.0 c	124.1 ± 2.4 b	86.4 ± 0.9 c	83.0 ± 2.4 f	25.6 ± 2.0 c	27.7 ± 1.1 d	3.4 ± 0.3 b	2.9 ± 0.1 c
	100 mM	31.7 ± 1.8 c	6.6 ± 1.0 a	31.8 ± 0.4 f	84.1 ± 2.2 e	54.6 ± 0.7 f	130.5 ± 2.1 a	136.4 ± 2.1 a	146.1 ± 1.4 a	0.4 ± 0.1 c	0.8 ± 0.1 d
HBL	0 mM	45.1 ± 2.4 ab	11.03 ± 2.1 a	49.9 ± 1.5 bc	127.8.8 ± 3.1 ab	88.9 ± 0.8 bc	86.5 ± 2.7 ef	23.7 ± 2.5 c	24.4 ± 1.4 de	3.8 ± 0.1 b	3.5 ± 0.2 b
	100 mM	35.3 ± 2.5 bc	7.1 ± 2.4 a	37.5 ± 1.0 e	94.4 ± 1.1 de	62.8 ± 1.2 e	121.0 ± 2.9 b	91.0 ± 2.0 b	124.1 ± 1.5 b	0.6 ± 0.1 c	0.9 ± 0.1 d
AMF	0 mM	48.9 ± 2.0 a	12.7 ± 1.4 a	51.6 ± 1.2 ab	132.8 ± 3.2 ab	91.8 ± 0.7 ab	89.5 ± 2.2 de	22.5 ± 2.7 c	22.8 ± 1.2 de	4.2 ± 0.4 ab	3.9 ± 0.1 b
	100 mM	41.7 ± 2.0 ab	8.1 ± 0.9 a	39.7 ± 0.9 de	105.4 ± 3.3 cd	67.2 ± 1.0 e	115.5 ± 1.8 b	87.9 ± 2.9 b	112.5 ± 1.5 c	0.7 ± 0.1 c	1.0 ± 0.2 d
HBL + AMF	0 mM	50.4 ± 1.7 a	13.4 ± 1.1 a	53.5 ± 0.7 a	139.1 ± 2.1 a	94.3 ± 0.4 a	94.8 ± 1.9 d	18.7 ± 2.6 c	20.4 ± 1.3 e	5.2 ± 03 a	4.6 ± 0.2 a
	100 mM	43.3 ± 2.2 ab	9.4 ± 1.9 a	41.4 ± 0.3 d	111.3 ± 1.8 c	72.8 ± 0.9 d	108.2±1.4 c	85.2 ± 2.3 b	107.4 ± 1.6 e	0.8 ± 0.2 c	1.0 ± 0.1 d

Data were collected for cucumber genotypes (Jinyou 1# and CCMC) 40 days (i.e., fruiting stage) after treatment with NaCl. Values presented for comparison are means of three replications in each treatment (±SE, standard error). Means followed by the same letters are not significantly different *(p ≤* 0.05) according to the least significant difference (LSD) having three replications.

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
