# Peer review of "The Protective Role of 28-Homobrassinolide and Glomus versiforme in Cucumber to Withstand Saline Stress"

_plants, 2019, doi:10.3390/plants9010042_

Round 1
Reviewer 1 Report
In this manuscript, Ahmad and collaborators study the protective role of 28-Homobrassinolide (HBL) and the arbuscular mycorrhizal fungi (AMF) Glomus versiforme individually applied or combined, on two different cultivars of cucumber exposed to salt stress (100 mM NaCl). With this aim, the authors measured several parameters such as fresh and dry weights, root length, chlorophyll content, electrolyte leakage, leaf relative water content, photosynthesis, colonization percentage, nutrient accumulation and the activity of different enzymatic antioxidants. They conclude that the use of HBL and AMF can be useful to improve plant growth, biomass and nutrition under saline conditions.
The manuscript is well written (though there are several typos) and can help further studies to deepen the role of AMF and HBL in plant protection under salt stress. Their results support and extent those previously obtained by other authors reporting that brassinosteroids (Marakli et al., 2015; Talaat et al., 2013; Rajewska et al., 2016) or AMF (Hashem et al., 2016; Schweiger et al., 2014; Zuccarini et al., 2008) treatments help plants to withstand salinity conditions. Nevertheless, in the opinion of this reviewer, there are several aspects of the manuscript and statements made by the authors that must be clarified for the paper to be suitable for publication.
Major concerns
In the Discussion of the manuscript, it is stated several times that cultivar Jinyou 1 # exhibits a lower tolerance to salinity than cultivar CCMC. This claim is based on the results from measurements of different parameters such as shoot and root length, fresh weight, chlorophyll content and photosynthesis, leaf relative water, electrolyte leakage or the activity of the different antioxidant enzymes. However, this can be hardly concluded from direct comparison of Jinyou 1 # and CCMC values in control vs. stressed plants (100 mM NaCl) shown in Tables 1-5 or Figs. 1-2. Supporting this claim requires determining the percentages of reduction in the parameter tested in stressed vs control plants for Jinyou 1 # and CCMC per treatment.
In the Results section, the authors should explain the reasons for using 28-Homobrassinolide and Glomus versiforme rather than other brassinosteroids or AM fungi. The same can be applied to the use of the Jinyou 1 # and CCMC cultivars.
The experimental procedures require clarification:
The number of samples and plants studied in each treatment is not clear. In Tables and Figs. legends it can be read that “Values are means of at least three replications” and in Materials and methods that each treatment was “replicated three times having ten pots per treatment”. This is confusing. Are the values of Tables and Figs. the mean of three different experiments or the mean of only one experiment with three different samples of ten pots each per treatment? The total number of plants studied per cultivar (n) and condition should be indicated.
Besides, the age of plants at each treatment should be shown in the legend of Tables and Figs.
Lines 110-112. The sentence “The HBL foliar application and AMF root inoculation alone and specifically their combinations revealed significant results in both cultivars at the respective periods mentioned above” requires clarification. The effect of individual and combined treatments should be explained more in detail.
Lines 120-121. “The HBL foliar application and AMF inoculation in roots (Figure 2) reduced the effects of stress”. This is not correct for photosynthesis and stomatal conductance because according to Fig.2 differences are not statistically significant for Jinjou1# and CCMC plants under 100 mM NaCl alone compared with 100 mM NaCl + HBL or 100 mM NaCl +AMF.
Colonization results (page 6). The effect of AMF + HBL in the presence of 100 mM NaCl is not relevant because AMF + HBL also caused a similar colonization increase in plants grown in the absence of salt stress.
I recommend summarizing and clarifying better in Discussion, under which of the studied conditions the effect of HBL + AMF over the individual treatments of HBL or AMF, represents a significant improvement on cucumber salt tolerance.
Minor concerns
- lines 61 and 73. Full names for AM and HBL (first time they are mentioned).
- lines 66-67. Full stop after rhizosphere. “hence are” should be replaced with “Hence AM fungi are directly related…”.
- Lines 82-85. “The application of HBL and AMF improved plant growth attributes and biomass under NaCl, however, the combined effect of HBL and AMF was observed as a significant increase in the shoot and root length, fresh and dry weight in cultivar Jinyou 1# and shoot 84 length, fresh and root dry weight in CCMC”. I suggest replacing it with this sentence: “The application of HBL and AMF improved plant growth attributes and biomass under NaCl. Besides, the combined effect of HBL and AMF caused a significant increase in the shoot and root length, fresh and dry weight in cultivar Jinyou 1# and shoot length, fresh and root dry weight in CCMC”.
- line 148. What is “inclined SOD”?
- Lines 157-159 and 175-177. The meaning of the text should be revised. What does “after ten days of cucumber” mean?
- Lines 171-173. Indicate that these differences were not statistically significant except for CCMC at 20 and 30 days.
- Table 5. Nitrogen root column. Letters are missing.
- Discussion. What does “significantly deteriorated damage” mean?
- Replace Stomata with stomata in some places of the manuscript.
- Table 5 header. Replace Shoot with shoot.
Author Response
We sincerely appreciate your critical and in-depth analysis of our article and are really thankful to you for your valuable suggestions which indeed helped us understand our mistakes and thereof improve our article. After careful reading of your comments, we found that the initial submission held lack of valuable information which might lead to misunderstanding our presentations. Thanks again for your suggestions through which, we could revise and improve our paper.
We hope that you will kindly consider our efforts and will provide us the opportunity to consider our revised article for acceptance.
The changes in the manuscript are highlighted with green color.
Question: In the Discussion of the manuscript, it is stated several times that cultivar Jinyou 1 # exhibits a lower tolerance to salinity than cultivar CCMC. This claim is based on the results from measurements of different parameters such as shoot and root length, fresh weight, chlorophyll content and photosynthesis, leaf relative water, electrolyte leakage or the activity of the different antioxidant enzymes. However, this can be hardly concluded from direct comparison of Jinyou 1 # and CCMC values in control vs. stressed plants (100 mM NaCl) shown in Tables 1-5 or Figs. 1-2. Supporting this claim requires determining the percentages of reduction in the parameter tested in stressed vs control plants for Jinyou 1 # and CCMC per treatment.
Answer:
We appreciate your in-depth analysis of our manuscript. After carefully reading your comment, we noticed that the confusion perhaps happened due to the unsuitable presentation design of the data tables. We revised the presentation style which now clearly indicates the differences between these two cultivars based on the observed parameters data. The interpreted results are duly mentioned in the results section in terms of percentages as you suggested. We hope you may find our revised data presentation appropriate to support our drafted results. Some of the examples are given for your kind consideration as follows:
Line 88-90: …shoot and root length (68 and 65%), fresh (64 and 41%) and dry weight (50 and 23%) in cultivar Jinyou 1#.........
Line 105-106: …… chlorophyll a (75%), b (54%), a+b (50%)…. cultivar CCMC as compared to control.
Lines 117-119: cucumber leaves after 20 days by 174 and 135 % in Jinyou 1# and CCMC, respectively.
Line 123-124: …stress by 76 and 56 % in cultivar Jinyou 1# while, 72 and 52 % in cultivar CCMC.
Line 125-126: … increased LRWC after 20 and 40 days by 28 and 29 % in cultivar Jinyou 1# while, 34 and 41 % in cultivar CCMC.
Line 133-134: NaCl decreased photosynthetic activity by 58 %, transpiration rate (38 %), …
Line 148: … decreased by 51% and 46% in cultivar Jinyou 1# and CCMC respectively.
Line 150: …increment in colonization percentage by 10% and 15% compared to stressed plants in both cultivars.
Line 166-167: …the activity of SOD after every 10 days by 20, 60, 61 and 72 %.
Line 173-174: …POD activity by 65, 71, 65, and 66 % after….
Line 193-194: … by 5, 18, 9 and 10 % in Jinyou 1# and….
Line 203-204: … concentration in shoots by 59 and 70 % in cultivar…
Line 205-207: … shoots (64 and 66 %) and roots (85 and 67 %) in…
Question: In the Results section, the authors should explain the reasons for using 28-Homobrassinolide and Glomus versiforme rather than other brassinosteroids or AM fungi. The same can be applied to the use of the Jinyou 1 # and CCMC cultivars.
Answer: We would like to clarify the reason behind selecting 28-Homobrassinolide and Glomus versiforme as the treatment specimens as follows:
The present work is part of on-going projects in our research groups which involves using Homobrassinolide and arbuscular micorhizal fungi to combat abiotic stresses such as heat, drought and salt stresses. Before this work, researches were conducted using 24-Epibrassinolide and Glomus etinucatum under salt stress and heat stresses. However, the experimental cost of 24-Epibrassinolide was much higher compared to the cost of 28-Homobrassinolide. Therefore, we selected this group of Brassinosteriods in order to relieve economic cost burden. In the meantime, the available species of AMF was Glomus versiforme in our lab research group so we conducted experiments by including these two treatments (HBL and AMF). Selection of cucumber CV. Jinyou 1# and CCMC was based on their reported tolerance levels to the salinity stress in Chinese literature. We therefore selected these two cultivars in order to justify the role of the applied treatments in ameliorating saline stress conditions on cucumber plants from two different genetic origins.
Agreeing to your suggestion, we have added the following sentences in the result section
Line 78-86: The time this research was started, according to our knowledge, no specific work showed the optimum dose of HBL to be substantial at a particular stress level of NaCl, therefore, we designed the experiments to work out and understand the considerate amount of HBL at variable stress levels of NaCl [36]. The AM Glomus species have been known to be widely distributed (regardless of the type and intensity of disturbance) in the ecosystem. Moreover, the versiforme specie was already inoculated and has been studied extensively at the college of horticulture and forestry (NWAFU). As stated, less was known about the role of HBL in ameliorating the salinity effects on growth of cucumber. Therefore, two different cultivars of cucumber were used in our study to elaborate the effectiveness of HBL per se in the salinity stress situations.
Question: The experimental procedures require clarification:
The number of samples and plants studied in each treatment is not clear. In Tables and Figs. legends it can be read that “Values are means of at least three replications” and in Materials and methods that each treatment was “replicated three times having ten pots per treatment”. This is confusing. Are the values of Tables and Figs. the mean of three different experiments or the mean of only one experiment with three different samples of ten pots each per treatment? The total number of plants studied per cultivar (n) and condition should be indicated.
Answer: We apologize for the lack of proper information which created confusions for the studied parameters data. Data were collected by selecting three random plant samples from each of the three experimental repeats. The means of these samples were further used to apply analysis of variance (ANOVA). We have carefully revised the tables and figures legends and added this information in order to avoid confusions for viewers.
Question: Besides, the age of plants at each treatment should be shown in the legend of Tables and Figs.
Answer: Thank you for your suggestion. We have found that your suggestion is of great value in order to improve the presentation of our paper and therefore have now mentioned the information regarding the age of cucumber plants in number of days counted after the application of NaCl treatments. The corrections made in the manuscript are as follows
Table 1: …. determined after 40 days of NaCl treatment, fruiting stage…
Table 2: … determined after 40 days of NaCl treatment, fruiting stage…
Table 3: … after 10, 20, 30 and 40 days (vegetative, flowering and fruiting stage) of NaCl…
Table 4: … after 10, 20, 30 and 40 days (vegetative, flowering and fruiting stage) of NaCl…
Table 5: …determined after 40 days (fruiting stage) of NaCl…
Figure 1: …after 20 and 40 days (flowering and fruiting stage) of NaCl treatment…
Figure 2: … after 40 days NaCl treatment, fruiting stage…
Figure 3: … end of experiment i.e. after 40 days (fruiting stage) …
Question: Lines 110-112. The sentence “The HBL foliar application and AMF root inoculation alone and specifically their combinations revealed significant results in both cultivars at the respective periods mentioned above” requires clarification. The effect of individual and combined treatments should be explained more in detail.
Answer: Thank you for your kind revision of our manuscript. The sentences have been added in the results as follows
Line 123-126: The leaves relative water content showed a significant decline after 20 and 40 days of stress by 76 and 56 % in cultivar Jinyou 1# while, 72 and 52 % in cultivar CCMC (Figure 1). The HBL foliar application and AMF root inoculation improved LRWC in cucumber leaves, however, their combinations increased LRWC after 20 and 40 days by 28 and 29 % in cultivar Jinyou 1# while, 34 and 41 % in cultivar CCMC
Question: Lines 120-121. “The HBL foliar application and AMF inoculation in roots (Figure 2) reduced the effects of stress”. This is not correct for photosynthesis and stomatal conductance because according to Fig.2 differences are not statistically significant for Jinjou1# and CCMC plants under 100 mM NaCl alone compared with 100 mM NaCl + HBL or 100 mM NaCl +AMF.
Answer: According to our understanding, this issue may have aroused due to the presentation styles of the figures. We revised the figures style presenting the data for each treatment more clearly. Additionally, the data for each cucumber cultivar was analyzed separately which only showed the comparison of treated plants to their respective controls only. That is why the claimed statements were not supported by the figures in the initially submitted manuscript. This problem is however, solved to our understanding after revising the figures style to a more clear and appropriate form.
Question: Colonization results (page 6). The effect of AMF + HBL in the presence of 100 mM NaCl is not relevant because AMF + HBL also caused a similar colonization increase in plants grown in the absence of salt stress.
Answer: In a similar manner, this confusion also existed due to our poor presentation data design in the initially submitted paper. We have revised the figure style and now you may clearly understand the difference between colonization percentage in the plants treated with salt stress to those without salt treatment. We sincerely apologize that due to the improper design of data presentation, this problem occurred.
Question: I recommend summarizing and clarifying better in Discussion, under which of the studied conditions the effect of HBL + AMF over the individual treatments of HBL or AMF, represents a significant improvement on cucumber salt tolerance.
Answer: The suggestion indeed improved our discussion and we have incorporated the suggested changes in the discussion section as:
Discussion:
Line 330-333: The higher tolerance of cucumber plants treated with the combination of HBL and AMF can be attributed to the increase in colonization of AM fungi in roots, phosphorus and potassium concentration in shoots, higher photosynthetic, oxidative enzymatic activity (SOD, and POD), and lower, lipid peroxidation under salt stress.
Minor concerns
lines 61 and 73. Full names for AM and HBL (first time they are mentioned).We wrote the full names for AMF and HBL as follows
Now Line 60: AMF (Arbuscular mycorrhizal fungi)
Now Line 71: HBL (28-Homobrassinolide)
lines 66-67. Full stop after rhizosphere. “hence are” should be replaced with “Hence AM fungi are directly related…”.We have changed it according to your suggestions.
Lines 82-85. “The application of HBL and AMF improved plant growth attributes and biomass under NaCl, however, the combined effect of HBL and AMF was observed as a significant increase in the shoot and root length, fresh and dry weight in cultivar Jinyou 1# and shoot 84 length, fresh and root dry weight in CCMC”. I suggest replacing it with this sentence: “The application of HBL and AMF improved plant growth attributes and biomass under NaCl. Besides, the combined effect of HBL and AMF caused a significant increase in the shoot and root length, fresh and dry weight in cultivar Jinyou 1# and shoot length, fresh and root dry weight in CCMC”.We have changed this sentence according to your suggestion as follows
Now Line 90-93: The application of HBL and AMF improved…. fresh and root dry weight in CCMC.
line 148. What is “inclined SOD”?Now Line 164: We apologize for the typing mistake. We have edited it and changed it to increase.
Lines 157-159 and 175-177. The meaning of the text should be revised. What does “after ten days of cucumber” mean?We have remove the mistake as follows
Now line 177-178 and line 195-198: after 10, 20, 30 and 40 days of NaCl
Lines 171-173. Indicate that these differences were not statistically significant except for CCMC at 20 and 30 days.We have added this sentence as follows
Now line 194: These differences were not statistically significant except for cultivar CCMC after 20 and 30 days of NaCl.
Table 5. Nitrogen root column. Letters are missing.Thank you for helping us to improve the quality of our manuscript, we have now provided the letters on nitrogen column.
8 Discussion. What does “significantly deteriorated damage” mean?
We have rephrased it as follows “lowers the damage”
Replace Stomata with stomata in some places of the manuscript.Thank you for your keen observation in improving the quality of our manuscript. We have replaced Stomata with stomata.
Line 69, 249, and 395.
Table 5 header. Replace Shoot with shoot.We have replaced the Shoot with shoot in the header of table 5.
Reviewer 2 Report
As I started to review this article, I found an article which was formerly published in Ecology and Evolution (Wiley), which is described very similar study:
The combination of arbuscular mycorrhizal fungi inoculation (Glomus versiforme) and 28-homobrassinolide spraying intervals improves growth by enhancing photosynthesis, nutrient absorption, and antioxidant system in cucumber (Cucumis sativus L.) under salinity
Ahmad et al. 2018, doi: 10.1002/ece3.4112
In my opinion, this article should be rejected.
Author Response
As I started to review this article, I found an article which was formerly published in Ecology and Evolution (Wiley), which is described very similar study:
The combination of arbuscular mycorrhizal fungi inoculation (Glomus versiforme) and 28-homobrassinolide spraying intervals improves growth by enhancing photosynthesis, nutrient absorption, and antioxidant system in cucumber (Cucumis sativus L.) under salinity
Ahmad et al. 2018, doi: 10.1002/ece3.4112
In my opinion, this article should be rejected.
Answer:
It is understandable that due to previous published work, current manuscript may have been mistaken as repeat. However, we would like to clarify our present work for its novelty as follows:
In our previous work, we presented the effects of 28-Homobrassinolide (HBL) spraying intervals combined with AMF root colonization to enhance cucumber plants tolerance to salinity stress. Focus was drawn on the application of HBL for 15- and 30 days intervals throughout cucumber growth however, this proved to be laborious and economically expensive in terms of farmers’ interests. We also noticed that increased application of HBL did not necessarily increased the root colonization of AMF in the treated plants. Therefore, we performed another experiment to optimize the minimum doze of HBL in terms of application that would successfully ameliorate the stress conditions on cucumber plants. Therefore, current manuscript presents the results obtained from a single application of HBL on cucumber with addition of AMF in the roots and then the plants were maintained throughout the growth period under saline stress and their growth and yield potential were recorded. The results therefore are of significance as a single application of HBL substantially improved the plant growth and these plants successfully survived the stress conditions. Moreover, a single application of HBL has less labor and economic cost. Hence both scientifically and economically valid, the present work maybe of importance to the readers and we hope you would kindly consider our efforts and give us a chance to publish our work.
Reviewer 3 Report
This paper is well written, although some minor grammatical errors need to be fixed before resubmission. However, I have one major and few general comments.
Major comment
In the present study, only two cucumber cultivars (salt sensitive vs tolerant) were used as such no much genetic diversity with regard to salinity tolerance and reaction upon application of phytohormones was captured. However, this study could have been improved by including more genetically diverse cucumber lines so that the findings should have a broader application. Generally, the effect of HBL and AMF on plant growth attributes under saline conditions may vary depending on the genetic background of cucumber cultivars. I understand this concern cannot be addressed now. In this case, I would suggest the authors to modify last sentence of the abstract, but rather I would say that these results are promising but further verification/study is needed.
General comments
Is there any specific reason why the authors used only two NaCl concentrations (i.e 0 and 100mM)?. It would have been better if variable rates of concentrations were used. Please include the analysis of variance (ANOVA) table as supplementary. Lines 84 and 85 indicate that combined effect of HBL and AMF under saline conditions was significantly higher (on plant growth attributes and biomass) than individual effect of HBL or AMF for CCMC, which is not true according to the results on Table 1. Table 1 shows that shoot length under AMF application was not significantly different from that under combined HBL and AMF. Similarly for root fresh and dry weights. This sentence should be rephrased. Similarly, on lines 95 and 96 for CCMC, in comparison with individual HBL and AMF, combined effect HBL and AMF was not significantly better on plant growth attributes (under saline conditions)-Table 2. Line 123 is not clear, beginning from…..while, photosynthesis,…….. The authors should re-write the last part of the sentence for clarity. Tables 3-5, are very busy. Can the authors present the results using figures as done in Figures 1 and 2?. Did the authors observe any significant effect of salinity on leaf number and area of the two cultivars?. Leaf number and area data should have been collected as well. In this study a lot of information was generated, however, I would suggest to include some of the results as Supplementary. In that way, the authors can expand the result/discussion points of the most important information.This manuscript has merit to be published in MDPI after minor revisions.
Author Response
We sincerely appreciate your critical and in-depth analysis of our article and are really thankful to you for your valuable suggestions which indeed helped us understand our mistakes and thereof improve our article. After careful reading of your comments, we found that the initial submission held lack of valuable information which might lead to misunderstanding our presentations. Thanks again for your suggestions through which, we could revise and improve our paper. After revising according to your kind suggestion, we feel our paper has improved and could meet the standard presentation required for a scientific article for publication.
We hope that you will kindly consider our efforts and will provide us the opportunity to consider our revised article for acceptance.
The Changes in the manuscript are highlighted with green color.
Comments
This paper is well written, although some minor grammatical errors need to be fixed before resubmission. However, I have one major and few general comments.
Major comment
Question: In the present study, only two cucumber cultivars (salt sensitive vs tolerant) were used as such no much genetic diversity with regard to salinity tolerance and reaction upon application of phytohormones was captured. However, this study could have been improved by including more genetically diverse cucumber lines so that the findings should have a broader application. Generally, the effect of HBL and AMF on plant growth attributes under saline conditions may vary depending on the genetic background of cucumber cultivars. I understand this concern cannot be addressed now. In this case, I would suggest the authors to modify last sentence of the abstract, but rather I would say that these results are promising but further verification/study is needed.
Answer: We cordially appreciate your scientific suggestion. At current, as you may agree, including in-depth research work is impossible however, we indeed will consider studying phytohormones and genetic backgrounds of the plants in our further research experiments. We have modified the sentence in the abstract according to your kind suggestion as:
Now Line 27-29: These results are promising, but further studies are needed to verify the crop tolerance to salinity and help in sustainable agricultural production, particularly vegetables that are prone to salinity
General comments
Question: Is there any specific reason why the authors used only two NaCl concentrations (i.e 0 and 100mM)?. It would have been better if variable rates of concentrations were used.
Answer:. We appreciate your concern, these salt conc. were selected were based on our previous research work (Canadian journal of plant science, https://doi.org/10.1139/cjps-2016-0404). However, for the convenience of the readers, we have now already stated this sentence in the M&M section.
The sentence is as follows
The selected treatments of NaCl were selected based on our previous research (Ahmad et al., 2018).
Question: Please include the analysis of variance (ANOVA) table as supplementary.
Answer: Thank you for your kind suggestion. We have provided the ANOVA table in the supplementary file.
Line 436-438:
Supplementary file: Table S1a to S44a presents ANOVA table for cultivar Jinyou 1#, and Table S1b to S44b for the cultivar CCMC.
Question: Lines 84 and 85 indicate that combined effect of HBL and AMF under saline conditions was significantly higher (on plant growth attributes and biomass) than individual effect of HBL or AMF for CCMC, which is not true according to the results on Table 1. Table 1 shows that shoot length under AMF application was not significantly different from that under combined HBL and AMF. Similarly for root fresh and dry weights. This sentence should be rephrased.
Answer: Thank you for your in debth analysis of our research work. We have rephrased the sentence in the manuscript as follows
Line 90-93---: The application of HBL and AMF improved growth attributes under salt stress….
Question: Similarly, on lines 95 and 96 for CCMC, in comparison with individual HBL and AMF, combined effect HBL and AMF was not significantly better on plant growth attributes (under saline conditions)-Table 2.
Answer: we have rephrased the sentences as follows
Now line 108-109: The combined effect of HBL and AMF showed an increase in chlorophyll a, b, total chlorophyll and root activity in Jinyou 1#, while the increase was not significant in cultivar CCMC.
Question: Line 123 is not clear, beginning from…..while, photosynthesis,…….. The authors should re-write the last part of the sentence for clarity.
Answer: Thank you for your valuable comment, we have rephrased the last sentences and are as follows
Line 133-136: …… the combined effect of HBL and AMF showed higher results in the increased intercellular carbon dioxide concentration, transpiration rate, photosynthesis and stomatal conductance in both cultivars under stress conditions
Question: Tables 3-5, are very busy. Can the authors present the results using figures as done in Figures 1 and 2?.
Answer: Your suggestion indeed helped us identify the lack of presentation in the tables and figures and therefore revised the tables by allowing a simple and plain explanation of the data. The revised tables now clearly define the treatments effects and we hope you may find them appropriate in the revised form. Presentation of the data in figure form however, was given up based on the overwhelming number of figures in the manuscript. We hope you will kindly agree to our presentation in the table form instead.
Question: Did the authors observe any significant effect of salinity on leaf number and area of the two cultivars?. Leaf number and area data should have been collected as well. In this study a lot of information was generated, however, I would suggest to include some of the results as Supplementary. In that way, the authors can expand the result/discussion points of the most important information.
Answer: We appreciate your scientific evaluation of our work. We tried to shift some of the data to the supplementary section, but the parameters mentioned are important for the better presentation of our research work. We would be thankful to you if you can consider our point of view.
Studying the effect of salinity on leaf number and area is indeed a considerable aspect of current research work. However, we would like to explain that in this current work, our major concern was to evaluate the role of HBL in colonization of AMF and the antioxidant defense system under prolonged salt stress. Therefore, we really hope that current study which is the part of our research project should be considered enough in the current status for acceptance. We hope you will kindly consider our point of view and provide us a chance to publish our work in the current status.
This manuscript has merit to be published in MDPI after minor revisions.
We are very thankful for the scientific evaluation of our manuscript.
Reviewer 4 Report
Comments for reviewers
The manuscript by Ahmad et al. entitled “The protective role of 28-Homobrassinolide and Glomus versiforme in cucumber cultivars to withstand saline stress involves antioxidant enzymes modulation and mineral uptake regulation” reports on the combined effect of arbuscular mycorrhizal fungi and homobrassinolide on salt stress tolerance in two cucumber cultivars. Other groups have reported that both AMF and HBL improve salt tolerance. What is new in this manuscript is that the combined effect of AMF and HBL on salt stress was tested in a different plant species. They analyzed various parameters relevant to salt stress response. Most data presented in tables and figures of most of the parameters support that the combined treatment improves salt tolerance. However, there are major concerns that need to be addressed before this manuscript is accepted for publication in this journal.
Major concerns
Pictures showing plant performance under the treatments is not provided to give visual impression Data on root K+ concentration is not consistent with salt tolerance phenotype. The treatments were expected to increase the concentration as compared to NaCl treatment. Similar to K+, the Na+ concertation also decreases in the root. Usually the two cations are taken up by the same mechanism, and with a decrease in Na+ uptake, an increase in K+ concentration is expected. The effect of the treatments on enzyme activity present in Table 4 is not very different between treatments. On page 2, Paragraph 2, it is stated that “The increase in antioxidant activity and a decrease in lipid peroxidation can be attributed to the improved mechanism of removing excess Na+ form plant cells and higher scavenging of reactive oxygen species.” This is not accurate. How does it improve removal of excess Na+ form plant cells.Others
The title is too long and needs to be shortened There are some typos in M & M. ‘cuumber’ instead of cucumber, ‘plant seeds’ instead of seeds, ‘and1 ml’ instead of and 1 ml In Result section, ‘AMF significantly inclined’…increased or enhanced? The conclusion may follow the discussion section.Author Response
We sincerely appreciate your critical and in-depth analysis of our article and are really thankful to you for your valuable suggestions which indeed helped us understand our mistakes and thereof improve our article. After careful reading of your comments, we found that the initial submission held lack of valuable information which might lead to misunderstanding our presentations. Thanks again for your suggestions through which, we could revise and improve our paper. After revising according to your kind suggestion, we feel our paper has improved and could meet the standard presentation required for a scientific article for publication.
We hope that you will kindly consider our efforts and will provide us the opportunity to consider our revised article for acceptance.
Comments for reviewers
The manuscript by Ahmad et al. entitled “The protective role of 28-Homobrassinolide and Glomus versiforme in cucumber cultivars to withstand saline stress involves antioxidant enzymes modulation and mineral uptake regulation” reports on the combined effect of arbuscular mycorrhizal fungi and homobrassinolide on salt stress tolerance in two cucumber cultivars. Other groups have reported that both AMF and HBL improve salt tolerance. What is new in this manuscript is that the combined effect of AMF and HBL on salt stress was tested in a different plant species. They analyzed various parameters relevant to salt stress response. Most data presented in tables and figures of most of the parameters support that the combined treatment improves salt tolerance. However, there are major concerns that need to be addressed before this manuscript is accepted for publication in this journal.
Major concerns
Question: Pictures showing plant performance under the treatments is not provided to give visual impression
Answer: We appreciate your suggestion which might be of good impression had quality pictures been taken in the first place. On the contrary, we could not provide visuals in the manuscript due to the fact that number of treatments, salt concentrations and genotypes altogether had an overwhelming effect and it was quite difficult to indicate the effects in a single picture. Therefore, we could not take quality pictures due to nearly impossible arrangement of all the treatments in one picture.
Question: Data on root K+ concentration is not consistent with salt tolerance phenotype. The treatments were expected to increase the concentration as compared to NaCl treatment. Similar to K+, the Na+ concertation also decreases in the root. Usually the two cations are taken up by the same mechanism, and with a decrease in Na+ uptake, an increase in K+ concentration is expected.
Answer: As stated, it is true the concentration of K+ was supposed to increase in case of a decrease in the Na+ concentration. The obtained data however showed otherwise. In our honest opinion perhaps it may be the result of experimental error or any flaw in the data analysis. But humbly request that this is the data we obtained and thereby presented in the manuscript.
Question: The effect of the treatments on enzyme activity present in Table 4 is not very different between treatments. On page 2, Paragraph 2, it is stated that “The increase in antioxidant activity and a decrease in lipid peroxidation can be attributed to the improved mechanism of removing excess Na+ form plant cells and higher scavenging of reactive oxygen species.” This is not accurate. How does it improve removal of excess Na+ form plant cells.
Answer: We apologize for the typo the paragraph. We have rephrased the sentence and it is now as follows
Line 284-286: The increase in antioxidant activity removes the excessively produced ROS which can be attributed to the lower lipid peroxidation in plants under stress.
Others
The title is too long and needs to be shortenedWe have changed the topic of our manuscript as follows:
The protective role of 28-Homobrassinolide and Glomus versiforme in cucumber to withstand saline stress
Instead of
The protective role of 28-Homobrassinolide and Glomus versiforme in cucumber cultivars to withstand saline stress involves antioxidant enzymes modulation and mineral uptake regulation
There are some typos in M & M. ‘cuumber’ instead of cucumber, ‘plant seeds’ instead of seeds, ‘and1 ml’ instead of and 1 ml In Result section, ‘AMF significantly inclined’…increased or enhanced?We have corrected the spelling mistake of cuumber to cucumber and seeds to plant seeds in line 352-353.
We have correct the typing mistake in and1 ml to and 1 ml.
Now Line 164: We apologize for the typing mistake. We have edited inclined SOD and changed it to increase.
The conclusion may follow the discussion section.
We have transferred the conclusion part to the end of the discussion.
Round 2
Reviewer 1 Report
The modifications made by the authors, especially the graphical changes, have substantially improved the manuscript. In general, most of the questions raised in my previous revision have been satisfactorily answered. However, there are still some inaccuracies (mostly typos) that must be corrected and some concerns that must be addressed prior publication.
Line 65. Move reference 23 before full stop and replace “hence” with “Hence”.
Line 79. Full stop after “NaCl”. Replace “therefore” with “Therefore”.
Line 80. Replace “considerate” with “adequate”, “suitable” or “appropriate”.
Line 123. Write (LRWC) after “The leaves relative water content”.
Line 126. Add this information: “Notwithstanding, the combined treatment significantly ameliorated LRWC only at 40 days in Jinyou 1”.
Line 142. Replace “Intercellular” with “intercellular”.
Lines 134-135. “The HBL foliar application and AMF inoculation in roots 134 (Figure 2) reduced the effects of stress”. I still do not fully agree with this statement. I suggest changing that sentence by this one: “The HBL foliar application and AMF inoculation in roots (Figure 2) reduced the effects of NaCl stress mainly in the CCMC cultivar and to a lesser extent in Jinyou 1”. This is because for Jinyou 1, AMF or HBL treatments in stressed plants do not result in significant differences compared with NaCl treated plants, for photosynthesis or stomata conductance.
Lines 147-150. This is correct, but HBL also ameliorates AMF colonization in the absence of NaCl. Hence, this is not a stress specific effect and HBL likely helps to cope with the stress by increasing AMF root colonization even without stress. This should be commented here or in the discussion.
Line 173. Correct “inclined POD activity”.
Lines 178 and 198. Replace “vetetative” with “vegetative”.
Line 214. Correct “genotypes”.
Author Response
Question: The modifications made by the authors, especially the graphical changes, have substantially improved the manuscript. In general, most of the questions raised in my previous revision have been satisfactorily answered. However, there are still some inaccuracies (mostly typos) that must be corrected and some concerns that must be addressed prior publication.
Line 65. Move reference 23 before full stop and replace “hence” with “Hence”.
Line 79. Full stop after “NaCl”. Replace “therefore” with “Therefore”.
Line 80. Replace “considerate” with “adequate”, “suitable” or “appropriate”.
Line 123. Write (LRWC) after “The leaves relative water content”.
Line 142. Replace “Intercellular” with “intercellular”.
Answer: Thank you for helping us to improve our manuscript. We have changed it in our manuscript according to your suggestions.
Queston: Line 126. Add this information: “Notwithstanding, the combined treatment significantly ameliorated LRWC only at 40 days in Jinyou 1”.
Answer: We have added this information accordingly now Line 132-133.
Question: Lines 134-135. “The HBL foliar application and AMF inoculation in roots 134 (Figure 2) reduced the effects of stress”. I still do not fully agree with this statement. I suggest changing that sentence by this one: “The HBL foliar application and AMF inoculation in roots (Figure 2) reduced the effects of NaCl stress mainly in the CCMC cultivar and to a lesser extent in Jinyou 1”. This is because for Jinyou 1, AMF or HBL treatments in stressed plants do not result in significant differences compared with NaCl treated plants, for photosynthesis or stomata conductance.
Answer: Thank you. We have changed it according to your suggestion.
Now line 141-143: The HBL foliar application and AMF inoculation in roots (Figure 2) reduced the effects of NaCl stress mainly in the CCMC cultivar and to a lesser extent in Jinyou 1#.
Question: Lines 147-150. This is correct, but HBL also ameliorates AMF colonization in the absence of NaCl. Hence, this is not a stress specific effect and HBL likely helps to cope with the stress by increasing AMF root colonization even without stress. This should be commented here or in the discussion.
Answer: Thank you for your kind suggestion. We have added this sentence in the discussion part.
Line 266-268: HBL also ameliorates AMF colonization in the absence of NaCl. Hence, this is not a stress specific effect and HBL likely helps to cope with the stress by increasing AMF root colonization even without stress.
Question: Line 173. Correct “inclined POD activity”.
Lines 178 and 198. Replace “vetetative” with “vegetative”.
Line 214. Correct “genotypes”.
Answer: We are really thankful to you for your valuable suggestions which indeed helped us understand our mistakes and thereof improve our article. We have incorporated the typos and suggestions according to your recommendation.
Reviewer 2 Report
This manuscript proved another evidence about protective effect of 28-homobrassinolide and Glomus versiforme in cucumber species during salt stress. Authors should write about Glomus versiforme species and their application as treatment on plants. Unfortunately, just in Results section we could read about this AMF. I think it would be much better to write about the two cucumber species in the Abstract part because this information about the selected cucumber species is needed to understand the main goals of this MS. The Results section began with a part like as an Introduction, so it needs to reconsider. The Chinese names of cultivars should be rewritten as sensitive or tolerant because it can help the readers to follow the data in the complex tables. I think authors should change the description and presentation of results. It is not clear that which days of treatment was examined, so I recommend to eliminate those days which we cannot see any relevant data. Tables are very confuse, so maybe it would be better to make bar diagrams instead of them. Discussion should be rewrite more focused mode.
Author Response
This manuscript proved another evidence about protective effect of 28-homobrassinolide and Glomus versiforme in cucumber species during salt stress.
Question: Authors should write about Glomus versiforme species and their application as treatment on plants. Unfortunately, just in Results section we could read about this AMF.
Answer: Thank you very much for valuable suggestion. We have added few sentences regarding the G. versiforme effect on plants under salt stress in the Introduction section.
The sentences can be seen in the manuscript as follows
Line 69-73: The density of AM spores in salt affected soils are reported to be low, however, [27,28] observed that the most predominant species of AMF in the severely saline soils of the Tabriz plains [with an electrical conductivity of around 160 dS m−1] were Glomus intraradices, G. versiform and G. etunicatum AMFG. versiforme is reported to make improvements in net photosynthetic rate, and water use efficiency [29].
Question: I think it would be much better to write about the two cucumber species in the Abstract part because this information about the selected cucumber species is needed to understand the main goals of this MS.
Answer: Thank you for your valuable suggestion. We have added the sentences regarding the two cucumber cultivars in the abstract section.
The sentences are as follows.
Line 17: …. cultivars (salt sensitive Jinyou 1# and tolerant, CCMC) .
Line 22:…in tolerant cultivar (CCMC) and to a lesser extent in sensitive (Jinyou 1#) cultivar.
Question: The Results section began with a part like as an Introduction, so it needs to reconsider.
Answer: according to your suggestion, we have removed the intro starting before our results section.
Question: The Chinese names of cultivars should be rewritten as sensitive or tolerant because it can help the readers to follow the data in the complex tables.
Answer: Thank you for your kind suggestion. We have added salt sensitive and salt tolerant in tables and figures in MS so that the reader can understand clearly.
Question: I think authors should change the description and presentation of results. It is not clear that which days of treatment was examined, so I recommend to eliminate those days which we cannot see any relevant data.
Answer: We have already changed the style of presentation of our results in the current MS according to the kind suggestions of reviewers. We tried our best to convey our message more clearly in the present form. I hope you will consider our efforts in this regard.
Question: Tables are very confuse, so maybe it would be better to make bar diagrams instead of them.
Answer: thank you very much for your kind suggestion. We have already tried to present our data in the form of figures, however, due to data in 4 different timings, and of two cultivars, it cannot be accommodated in a single figure. The only best way for us to present it in the form of table for clarity.
Question: Discussion should be rewrite more focused mode.
Answer: Thank you for your valuable suggestion. Up to our knowledge, we have tried our best to add, rewrite or rephrase the following sentences in order to further improve our discussion.
The sentences are as follows:
Line 233: …which causes disruption in cell organelles…
Line 234-235: …higher symbiosis was observed….
Line 248-249: …in turn, decreases photosynthetic activity and…
Line 266-267: HBL also ameliorates AMF colonization in…..
Line 296-298: The increase in antioxidant activity…
Line 303-304: SOD has an affinity…
l;ine 324-325: and disturbs ionic hemostasis…
We sincerely appreciate your critical and in-depth analysis of our article and are really thankful to you for your valuable suggestions which indeed helped us understand our mistakes and thereof improve our article. After revising according to your kind suggestion, we feel our paper has improved and could meet the standard presentation required for a scientific article for publication.
We hope that you will kindly consider our efforts in this regard.
Thank you
Round 3
Reviewer 2 Report
I accepted the responses of authors.